# Never a Dull Moment: Distributional Properties as a Baseline for Time-Series Classification

## Abstract

The variety of complex algorithmic approaches for tackling time-series classification problems has grown considerably over the past decades, including the development of sophisticated but challenging-to-interpret deep-learning-based methods. But without comparison to simpler methods, it can be difficult to determine whether such complexity is required to obtain strong performance on a given problem. Here, we evaluate the performance of an extremely simple classification approach: a linear classifier in the space of two basic features that ignore the sequential ordering of the data: the mean and standard deviation of time-series values. Across a large repository of 129 (after filtering) univariate time-series classification problems, this simple distributional moment-based approach outperformed chance on 71 problems and reached 100% accuracy on two problems. In an additional neuroimaging time-series classification case study, we find that a simple linear model based on the mean and standard deviation performs better at classifying individuals with schizophrenia than a model that additionally includes features of the time-series dynamics, with performance sitting within the range of current literature. We conclude that comparing the performance of simple distributional features of a time series provides important context for interpreting the performance of more complex features or methods, which may not always be required to obtain high accuracy.

## 1 Introduction

Time-series classification is a key problem in the sciences and industry, wherein time-varying data is used to distinguish labeled classes. The quantity and diversity of time-series classification algorithms is large and increasing, from simple linear decision boundaries in interpretable feature spaces (Fulcher & Jones, 2014) to complex methods based on deep neural networks, such as long short-term memory networks (Ismail Fawaz et al., 2019). Complex new algorithms can yield impressive classification accuracy on challenging problems, but often at the expense of clear human interpretability (Rudin, 2019).

The University of East Anglia/University of California Riverside (UEA/UCR) univariate time-series classification repository, which currently contains 143 problems spanning a variety of domains (Bagnall et al., 2022), has been crucial for encouraging transparent reporting of the relative strengths and weaknesses of classification algorithms. It has also enabled the field to overcome key limitations in prior standard practice of reporting and comparison, including avoiding cherry-picking of optimistic datasets when reporting algorithm performance (Keogh & Kasetty, 2003). Systematic comparisons of time-series classification algorithms across this database have been essential for benchmarking the accuracy of state-of-the-art algorithms (Bagnall et al., 2017; Middlehurst et al., 2024). However, in settings ranging from policy making (Goodman & Flaxman, 2017) to healthcare (Caruana et al., 2015), deriving an interpretable understanding that can guide subsequent decision making can be more important than absolute classification accuracy. In such settings, simpler methods that are faster to train and clearer to interpret are often preferred.

Recent work has shown that parsimonious and interpretable methods can match (or even outperform) more sophisticated ones, in settings ranging from sleep-stage classification (Van Der Donckt et al., 2023) to earthquake detection (Waheed et al., 2020). A particularly striking example was the demonstration of a two-

parameter logistic regression model with equivalent performance in earthquake aftershock forecasting to a deep neural network with thousands of parameters (Mignan & Broccardo, 2019). For such settings, in which simple methods can perform well, opting for more complex models risks over-fitting, leading to poor generalizability on out-of-sample data. This effect was recently demonstrated in a meta-analysis of neuroimaging-based classifiers for autism spectrum disorder, in which deep-learning models exhibited inferior performance to linear support vector machine (SVM) models on unseen data (Traut et al., 2022).

Defaulting to the use of complex classification models, and only comparing their performance to chance-level accuracy and that of other complex models, can thus result in classifiers that are over-complicated and difficult to interpret. This issue is well-illustrated by the time-series classification task of distinguishing 'epilepsy' from 'eyes-open' states from electroencephalogram (EEG) data (Andrzejak et al., 2001). While state-of-the-art approaches had applied complex algorithmic approaches, ranging from independent components analysis and discrete wavelet transforms to multi-layer neural networks (Subasi & Ismail Gursoy, 2010), it was found that the time-series standard deviation alone could almost perfectly separate the two classes (Fulcher et al., 2013). The strong performance of a simple threshold classifier on standard deviation suggests itself as a more interpretable and parsimonious alternative to complicated algorithmic approaches to this problem, highlighting the utility of starting with a simple model and building in complexity only when it yields clearly demonstrable benefits.

Given that the characteristic complexity of time-series classification tasks (relative to classifying non-sequential data) relates to the challenge of quantifying class-informative dynamical patterns, here we aimed to investigate the performance of an extremely simple benchmark: a linear classifier in the two-dimensional space of the mean and standard deviation (which ignore the sequential ordering of the data entirely). As these two features instead characterize the distribution of time-series values, strong performance of these simple benchmark features indicates problems for which trivial properties of the distribution are sufficient to perform well, undermining the need for more complex approaches that aim to quantify informative temporal patterns (Fulcher, 2018). While some previous work (Fulcher & Jones, 2014; Lubba et al., 2019) has $z$-scored all time series prior to analysis—thereby insuring mean and variance are uninformative, to focus on dynamical patterns—such normalization has not been applied consistently across all problems in the UEA/UCR Repository (Bagnall et al., 2017; Middlehurst et al., 2024). This means that there is a collection of problems in the repository for which approaches sensitive to simple distributional properties may perform well—but these results may not hold if normalization is uniformly applied. We also investigate the performance gains of additionally incorporating a simple set of time-series features that capture different types of dynamical patterns in the data, using the `catch22` feature set (Lubba et al., 2019). None of these features is sensitive to the mean or standard deviation of the data; they instead capture more subtle time-series properties, including aspects of distributional shape; periodic patterns; temporal spacing of outliers; and linear and nonlinear autocorrelation (Lubba et al., 2019). Finally, to emphasize the practical implications of our findings beyond time-series data in the UEA/UCR Repository, we compare how well the mean and variance of whole-brain resting-state activity can classify individuals with versus without schizophrenia.

## 2 Methods

This study aims to evaluate the performance of a simple baseline linear classifier for time series in the two-dimensional space of their mean ($\mu$) and standard deviation ($\sigma$): features related to the First Two Moments (FTM) of the distribution. We first discuss methods related to fitting the FTM and `catch22` feature-based time-series classifiers across the UEA/UCR Repository (Bagnall et al., 2022) (in Sec. 2.1) and then describe specific methods related to our neuroimaging case study (in Sec. 2.2).

### 2.1 Feature-based classifier performance on the UEA/UCR Repository

In the UEA/UCR Time-Series Classification Repository (Bagnall et al., 2022), each of the 143 problems (as downloaded on 16 April 2026) is partitioned into designated train and test sets. These problems vary widely in the number of time series, the relative size of train and test set sizes, time-series lengths, and number of classes [see (Bagnall et al., 2022; 2017; Middlehurst et al., 2024) for details]. For all time series, we computed the FTM feature set containing two features: the mean ($\mu$) and standard deviation ($\sigma$) Henderson & Fulcher

(2025). For comparison to more sophisticated features of the time-series dynamics, we used the `catch22` set of 22 time-series features which are sensitive to properties such as linear and nonlinear autocorrelation structure, extreme event timing, and long-range correlations (Lubba et al., 2019). Given a feature space, we fit and evaluated classification models following the resample-based procedure outlined in (Bagnall et al., 2017; Middlehurst et al., 2024), using 30 resamples of train–test splits. That is, in addition to the designated split, we generated 29 additional (seeded) resamples of the data that preserve the class proportions of the designated split, for a total of 30 train–test splits. Prior to fitting a linear SVM, features were normalized as a *z*-score by computing the mean and standard deviation of each feature in the train set, and using these values to rescale both the train and test sets (ensuring that test data was completely unseen). A linear SVM was fit on the train set and used to generate predictions for the test set. Classification accuracy was used as the performance metric, following prior work (Bagnall et al., 2017; Middlehurst et al., 2024).

We then performed two sets of comparative analyses: (i) a comparison of FTM performance relative to the chance probability for each problem; and (ii) a comparison of FTM to the union set of 24 features from both `catch22` and FTM (`catch22` + FTM). To determine the problems on which FTM beats chance for the first analysis, we calculated the *p*-value for the one-sided test of the chance probability under the empirical accuracy distribution of the 30 resampled FTM accuracy values:

$$p = 1 - \Phi\left(\frac{\mu - p_{\text{chance}}}{\sigma}\right) \tag{1}$$

where $\mu$ and $\sigma$ are the mean and standard deviation of classification accuracy values for FTM, $p_{\text{chance}}$ is the chance-level accuracy probability for the problem, and $\Phi(\cdot)$ is the standard normal cumulative density function. We used a threshold of $\alpha = 0.05$ on these *p*-values to indicate a statistical difference between FTM and chance.

To statistically compare the performance of FTM and the union set of `catch22` + FTM in the second analysis, we implemented the correlated test statistic (Nadeau & Bengio, 2003) that corrects the traditional *t*-statistic to account for the violation of the assumption of independence incurred in the usage of resampling. This statistic can be written as:

$$t = \frac{\frac{1}{n}\sum_{j=1}^{n} d_j}{\sqrt{(\frac{1}{n} + \frac{n_2}{n_1})\sigma^2}}, \tag{2}$$

where $d_j$ is the difference in classification accuracy between the two time-series feature sets for the *jth* resample, $n$ is the number of resamples, $n_1$ is the train test size, $n_2$ is the test set size, and $\sigma^2$ is the variance of all resampled classification accuracy values. The $(\frac{1}{n} + \frac{n_2}{n_1})$ scaling factor acts as an estimator for cross-sample variance. We used the *correctR* package for R to implement the calculation computationally (Henderson, 2022). Using this corrected *t*-statistic, we then computed a *p*-value for a one-way hypothesis test that the mean of the `catch22` + FTM feature set distribution was larger than the mean of the FTM distribution. We again used a threshold of $\alpha = 0.05$ to indicate that there was a statistical difference between the means.

Lastly, to ensure a fair comparison between the FTM and `catch22` + FTM feature sets, we removed fourteen problems (such as `AllGestureWiimoteX`, `AllGestureWiimoteY`, `AllGestureWiimoteX`, and `PLAID`) from all analyses due to the presence of time series that had $< 10$ real values—the minimum threshold for calculations in `catch22`. We analyze the remaining 129 problems throughout the rest of this paper.

## 2.2 Schizophrenia classification from fMRI

As a separate case study, we additionally investigated the performance of simple feature-based classifiers to distinguish adults with schizophrenia (SCZ) versus cognitively healthy controls based on their whole-brain activity dynamics (Bryant et al., 2024). We obtained resting-state functional magnetic resonance imaging (rs-fMRI) data from the University of California at Los Angeles Consortium for Neuropsychiatric Phenomics

LA5c Study (Poldrack et al., 2016), which was pre-processed using the ICA-AROMA + 2P + GMR method (Pruim et al., 2015), as described previously (Aquino et al., 2020). Blood oxygen level-dependent (BOLD) signals consisting of 152 time samples were extracted for each of 68 cortical (Desikan et al., 2006) and 14 subcortical (Fischl et al., 2002) regions per participant. We excluded $N = 5$ participants ($N = 3$ Control and $N = 2$ SCZ) with constant time series across all brain regions after preprocessing. For quality control, we further excluded $N = 2$ participants with a mean framewise displacement greater than 0.5mm as described in Parkes et al. (2018), yielding a final sample of $N = 164$ participants ($N = 116$ control and $N = 48$ SCZ). The control and SCZ groups differ in terms of age (Control $= 31.2 \pm 8.7$ years; SCZ $= 36.6 \pm 9.0$ years; Wilcoxon rank sum test $p = 7 \times 10^{-4}$) and sex (Control $= 46.6\%$ Female; SCZ $= 25\%$ Female; $X^2(1, N = 164) = 5.7$, $p = 0.01$).

After feature extraction using *theft* (Henderson & Fulcher, 2025), the dataset was in the form of an $N \times R \times F$ matrix, for $N = 166$ subjects, $R = 82$ brain regions, and $F$ features. We incorporated all combinations of 82 brain regions and features (either $F = 2$ FTM features or $F = 24$ FTM + `catch22` features) as inputs to a linear SVM classifier. In other words, the FTM-only model had a total of $2 \times 82 = 164$ SVM input features, and the FTM + `catch22` model had a total of $24 \times 82 = 1968$ features. Classifiers were implemented with the `SVC` function from *scikit-learn* (Pedregosa et al., 2011), setting the regularization parameter `C = 1`, disabling the shrinking heuristic, and using balanced class weights.

Model performance was evaluated as balanced accuracy measured from stratified 10-fold cross-validation (CV) with 10 repeats. For each $k$-fold, the same $z$-score feature-normalization was applied as described in Sec. 2.1 above. Balanced accuracy is reported as the mean $\pm$ SD across the 10 repeats, in which each repeat contains the mean balanced accuracy across the 10 CV folds. To quantify the difference in model performance with FTM alone versus with the inclusion of `catch22`, we applied the same corrected $T$-statistic as above, this time using a two-tailed test.

## 3 Results

Results are structured as follows. First, in Sec. 3.1, we describe findings across the UEA/UCR Repository, where we demonstrate that the FTM classifier outperforms chance on the majority of problems. We then highlight the practical ramifications of these results using the example of a neuroimaging biomarker classification task in Sec. 3.2, in which the mean and standard deviation exhibit strong performance that is weakened by adding dynamical properties of the functional neuroimaging time series.

### 3.1 Linear FTM classifier performance across the UEA/UCR Database

We first investigated how a linear classifier based on the two FTM features, $(\mu, \sigma)$, performed across the 129 problems from the UEA/UCR Repository. We found that the FTM-based classifier statistically outperformed chance ($p < 0.05$) on 71 of the 129 problems (55.0%). FTM-based classification accuracies relative to chance level are plotted for these 71 problems in Fig. 1. We note that chance is often beaten by a substantial margin on these problems. Indeed, this simple classifier achieves 100% accuracy on two problems: `InsectEPGRegularTrain` and `GunPointOldVersusYoung`. For the 52 problems for which the FTM-based classifier did not beat chance, such a finding could be due to two reasons: (i) the dataset has been correctly $z$-scored in the UEA/UCR Repository (meaning each time series has a mean of zero and standard deviation of one); or (ii) the FTM features are not informative of class differences for that problem. These results demonstrate that, even on a repository devoted to time-series classification tasks, strong performance can be obtained using the two simplest distributional statistics that are unrelated to the sequential ordering of the data, due to the database containing problems in which labeled classes have distinctive levels (means) or scales (variances).

To better understand how the FTM classifier is behaving, we examined the two problems for which it achieved 100% cross-validated accuracy. For `GunPointOldVersusYoung`, the classes are clearly distinguished in the $(\mu, \sigma)$ feature space (Fig. 2A), while for `InsectEPGRegularTrain` each class has a characteristic mean level (Fig. 2B). For the binary ('Young' vs 'Old') `GunPointOldVersusYoung` task, the time series is the $x$-axis coordinate of the centroid of the actor's right hand at each frame while moving it from rest position to a gun

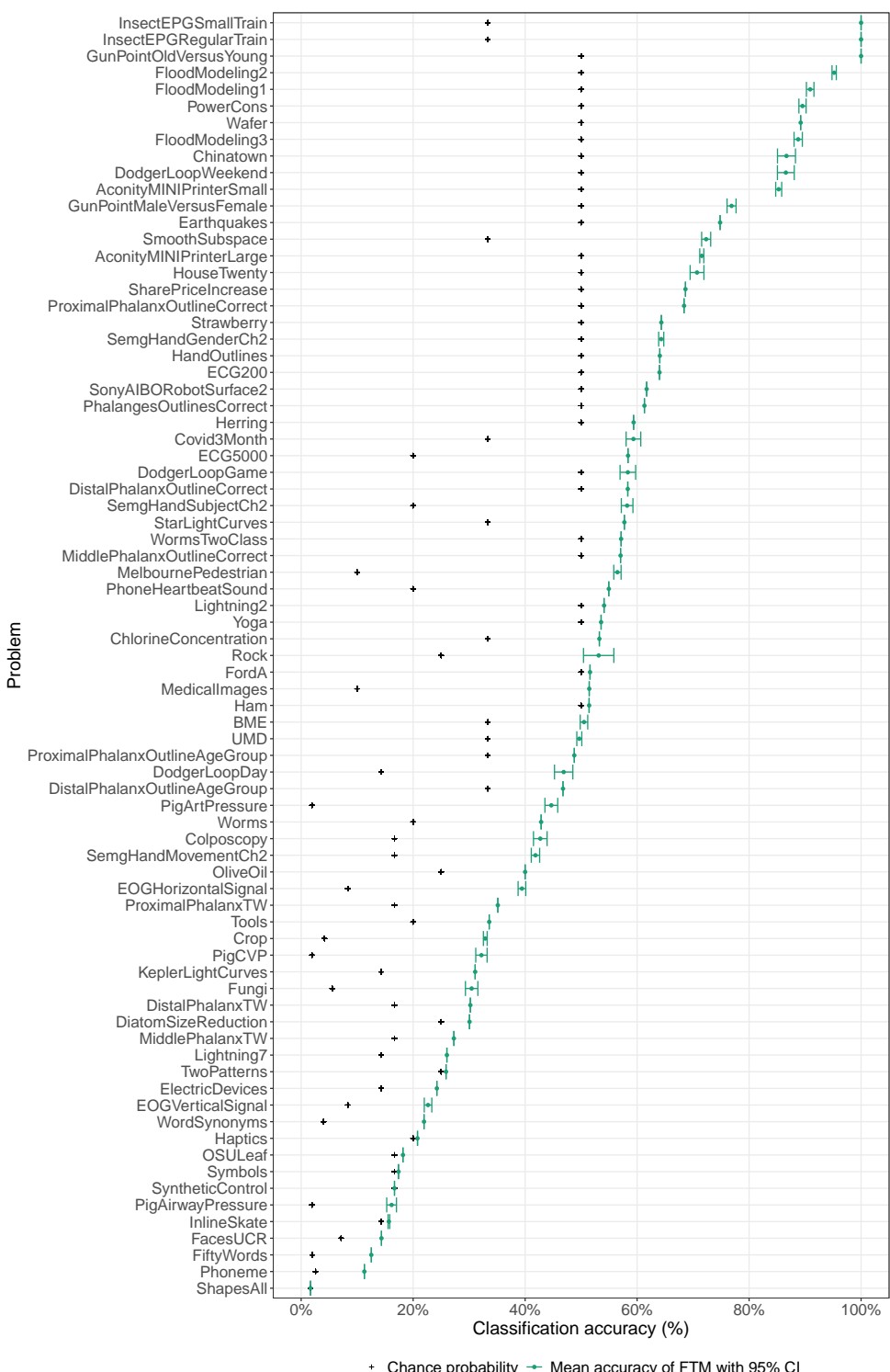

Figure 1: **Mean and standard deviation as features in a linear SVM classifier statistically outperforms chance-level accuracy on 71 of 129 problems in the UEA/UCR time-series repository.** Classification accuracy (%) is displayed along the horizontal axis and the name of each problem is shown on the vertical axis. Chance accuracies are displayed as black crosses. Points indicate mean classification accuracy and error bars show 95% confidence intervals (across train–test resamples) computed via the $t$-distribution.

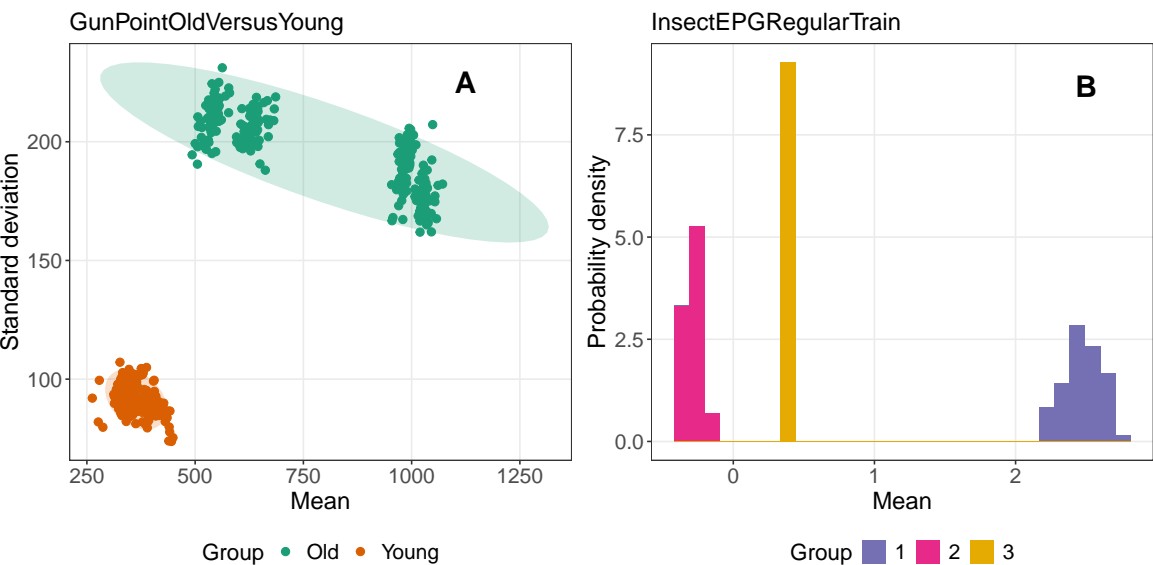

Figure 2: **Simple distributional moments can perfectly separate classes in the `GunPointOldVersusYoung` and `InsectEPGRegularTrain` datasets.** **A** Individual time series in `GunPointOldVersusYoung` are represented in the two-dimensional feature space of mean and standard deviation. Points are colored by class (as labeled), and class-level covariances are shown as shaded ellipses to guide the eye. **B** Histogram of time-series means by class in `InsectEPGRegularTrain`.

pose and back again (Ratanamahatana & Keogh, 2004). Figure 2A shows that the 'Young' actor exhibits characteristically lower mean and variation of their hand coordinate, consistent with a shorter height and arm length than that of the 'Old' actor. It is worth noting that, in contrast to features of the time-series dynamics (many of which are invariant to linear rescalings of the time-series values), $\mu$ and $\sigma$ are highly sensitive to the calibration of experimental measurements. Since small differences in the camera angle or distance from the actor can have a large effect on $\mu$ and $\sigma$—and thus on accurate classification—these features are less likely to generalize well to new data points for the `GunPoint` collection of problems (Wu et al., 2023). In `InsectEPGRegularTrain`, each class has a characteristic mean voltage level of the electrical circuit that connects insects with their food source, such that classes can be accurately distinguished based on their mean voltage without considering sequential patterns.

Finally, we tested our hypothesis that including dynamical features contained within `catch22` with the FTM distributional features would improve classification performance across time-series problems. We compared classification performance using the FTM feature set alone (2 features) versus the `catch22` + FTM feature set (24 features) across the 129 problems. Adding `catch22` features resulted in a mean absolute improvement in classification accuracy of 31.5% across the 129 problems, confirming the general importance of capturing dynamical properties for time-series classification problems. However, pairwise comparisons using the corrected test statistic revealed that there was no statistical difference between (*i*) FTM and (*ii*) `catch22` + FTM, on 22 of the 124 (or 17.1% of) problems. This demonstrates that, for many problems, simple distributional properties yield a surprisingly strong baseline against which to assess the benefits gained by using more complex approaches.

## 3.2 Neuroimaging Biomarker Case Study

We next extended our investigation of simple baseline time-series classifiers to neuropsychiatric disorder classification from rs-fMRI—a domain in which recent years have seen performance gains largely by combining opaque deep neural network methods with relatively small sample sizes (Quaak et al., 2021; Zeng et al., 2018; Ghanbari et al., 2023; Chen et al., 2023; Khullar et al., 2021; Subah et al., 2021; Ahmad et al., 2023; Raeisi

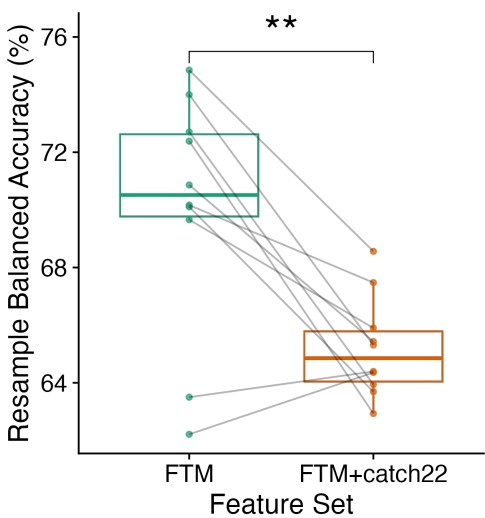

Figure 3: **The FTM feature set shows strong performance at classifying SCZ cases versus control participants from resting-state fMRI time series.** The distribution of mean balanced accuracy (across 10 repeats), using the combination of all 82 brain regions and either the two *FTM* features (green), or the 24 FTM + `catch22` features (orange), is shown as a combined spaghetti plot and boxplot. For each boxplot, the horizontal line within the box denotes the median and the upper and lower bounds reflect the first and third quartiles, respectively. Each gray diagonal line corresponds to one repeat out of ten, connecting the balanced accuracy results (computed across $k = 10$ folds) for the pair of feature sets along the $x$-axis. **: $p = 0.03$, corrected resampled $T = 2.64$.

et al., 2025). However, the combination of complex models with small sample sizes can yield over-fitted classifiers that do not generalize to unseen and/or held-out validation datasets (Flint et al., 2021; Winter et al., 2024; Belov et al., 2024; Itahashi et al., 2025). Given the strong performance of simple distributional statistics for some time-series classification problems above, we hypothesized that FTM might present a parsimonious pair of time-series features to distinguish individuals with schizophrenia (SCZ) from healthy controls. We therefore compared the performance of FTM alone versus with the inclusion of the full `catch22` feature set in distinguishing SCZ cases from controls based on rs-fMRI scans (see Methods, 2.2). As shown in Fig. 3, the classification model based on FTM features displayed a high balanced accuracy ($70.0 \pm 4.2\%$), sitting well within the range of a recent meta-analysis of dedicated SCZ classification studies using rs-fMRI (de Filippis et al., 2019). Adding `catch22` features to the model decreased its performance to $65.2 \pm 1.7\%$ (corrected resampled two-tailed $T$-statistic $= 2.64$, $p = 0.03$). Consistent with the findings on some problems in the UEA/UCR Repository above, these results demonstrate the surprisingly strong performance of basic properties of the distribution of time-series values for this fMRI classification task. It is further surprising that incorporating information about subtle patterns in this rich and complex whole-brain time-series dataset (using `catch22` features) yielded models with inferior performance to that of a simple linear classifier based on the mean and standard deviation alone.

## 4    Discussion

With the growing sophistication of machine-learning algorithms, modern data analysts are increasingly adopting methodological approaches that default to more complex statistical methods, which can obscure their clear interpretation. Presenting such complex approaches without direct comparison to simpler alternatives makes it difficult to discern whether the methodological complexity is beneficial, an assumption that has been challenged in some recent studies (Van Der Donckt et al., 2023; Waheed et al., 2020; Mignan & Broccardo, 2019). In this work, focusing on time-series classification problems, we demonstrate that a method at the extreme end of simplicity—using just two distributional-moment-based features, $\mu$ and $\sigma$, and

a linear classifier—performs surprisingly well on many problems. Despite being insensitive to the unique property of sequential data relative to an unordered vector—its ordering (in time)—this simple benchmark classifier statistically outperformed chance on approximately half of the problems in a prominent time-series classification archive, the UEA/UCR Repository (Bagnall et al., 2022), and even achieved 100% accuracy on two problems. Our results underscore the importance of carefully considering the factors that contribute to model performance, assessing model gains relative to simpler models, and favoring parsimony by carefully building up model complexity incrementally from simple and interpretable baselines.

We demonstrated the applicability of our results to functional neuroimaging-based classification of schizophrenia, a setting in which complex models are often implemented with high classification performance, though this may not generalize out of sample (Flint et al., 2021; Traut et al., 2022). Importantly, classification performance was stronger with FTM than with a model that also included `catch22` features of the dynamics using a linear SVM classifier. Findings presented in a separate analysis demonstrated that the standard deviation ($\sigma$) on its own—measured across all brain regions—distinguished SCZ cases from controls better than the mean ($\mu$) or any of the `catch22` features individually (Bryant et al., 2024). Here, we expand upon this to show that the FTM of resting-state activity measured across the brain outperformed a linear model that included all `catch22` features (which capture dynamical properties). In addition to genuine biological differences in the FTM of whole-brain activity in SCZ patients from controls, other potential factors could also contribute to its superior classification performance to that of the `catch22` features. For example, the smaller SVM feature space for FTM (164 features) compared with `catch22` (1968) is less susceptible to noise accumulation (Jamalabadi et al., 2016) that can adversely affect out-of-sample classification performance. Additionally, the relatively low temporal resolution inherent to fMRI (TR=2s for the dataset examined here) could obscure more nuanced nonlinear properties of neural activity that preclude the utility of the `catch22` features compared with properties like the mean and standard deviation that are insensitive to timepoint ordering.

The strong performance of mean and standard deviation across the UEA/UCR Repository demonstrates that time series in these problems have not been consistently normalized, as prior work has highlighted (Bagnall et al., 2017). If all time series were individually $z$-score transformed, there would be no class differences in mean or standard deviation for any problem (and the FTM-based classifier would exhibit null performance). Our results have implications for comparing time-series classification algorithms on problems for which labeled classes can be distinguished based on distributional properties alone. For example, consider the `catch22` features, which are all insensitive to the mean and variance of the input time series (and the time series are explicitly $z$-scored prior to feature computation) (Lubba et al., 2019). Relative to `catch22`, the superior performance of an alternative time-series classifier (which *is* sensitive to $\mu$ or $\sigma$ of the input series) could be driven entirely by a class-level difference in $\mu$ or $\sigma$—properties that are unrelated to dynamical patterns, and that `catch22` cannot access. One approach to test the ability of different classification algorithms to capture properties related to the dynamics of a uniformly sampled univariate time series would be to normalize time series such that there are no class differences in basic distributional properties. A second approach would be to compare model performance to a benchmark in which only simple distributional properties are included, with gains relative to this benchmark then being attributable to class-relevant differences in more complex properties. In this latter approach, there is scope for extending the two simple FTM features used here by adding higher-order moments, or other types of distributional features [such as those included in comprehensive time-series feature sets like *hctsa* (Fulcher & Jones, 2017) and *theft* (Henderson & Fulcher, 2025)].

This work also highlights the need for careful consideration of the generalizability of models that use features sensitive to the calibration of time-series measurements (like $\mu$ and $\sigma$), relative to features that are invariant to linear rescaling of the input time series (like the `catch22` feature set). This is because such measurement-scale dependent features are highly sensitive to changes in the calibration of experimental measurements that may not be precisely maintained in new data. An illustrative example is shown here in the `GunPointOldVersusYoung` dataset, for which our FTM-based method achieved 100% accuracy. However, this may be a deceptively high value, as it is reliant on the precise calibration of new data. A prior analysis of this dataset, focusing on early classification, has shown that classification accuracy can plummet with only slight variability in experimental calibration (to changes as small as a $\approx 1.9°$ tilt in the angle of the

camera used to measure hand coordinates) (Wu et al., 2023). In general, the decision of whether or not to include features that are sensitive to measurement scale (or to focus on scale-invariant features of the dynamics or distribution shape, as in `catch22`) should be motivated by domain expertise to avoid overly optimistic classification results.

In summary, by highlighting many problems for which simple distributional properties of a time series can achieve surprisingly high classification accuracy, our results raise important issues for the development and interpretation of time-series classification models. Future work evaluating time-series classification algorithms may consider using simple benchmarks for comparison to aid interpretation and provide evidence for the contribution of model complexity to any performance advantage, particularly for problems highlighted here for which simple distributional features are highly informative of class differences.

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
