# OpenReview forum: "Never a Dull Moment: Distributional Properties as a Baseline for Time-Series Classification"
_TMLR — Under review for TMLR_

### Review · Reviewer_Z4r6 · 2026-05-28

**Summary Of Contributions:**

The authors analyze time series data using a simple SVM based method, starting with two simple features: the mean and variance across the time series. On the UEA/UCR univariate time-series classification repository, they find that these two simple features do better than chance in a majority of cases. Adding features from the catch 22 feature set, which includes time-aware features can improve the performance.

The authors also apply the method to an FMRI task, where the mean and variance based feature method outperforms both the full catch 22 feature set but also a neural network based model. They conclude that there are classification tasks where simple summary statistics can be used to construct useful classifiers.

**Audience:**

No

**Audience Explanation:**

While I believe the paper is well written and well presented, it's not clear to me that the results will be interesting for people working in time series analysis. The fact that FTM classifies better than chance is not necessarily surprising; there are also domains in which it will likely fail badly (as the authors themselves acknowledge). The comparison with the catch 22 features is not surprising as well. The most interesting part perhaps is the comparison of the SVM model to the neural network model on the schizophrenia dataset, but this is a one-off experiment and it's hard to draw conclusions from it.

I'm curious what the other reviewers think on this point; I have not worked much in time series analysis so I don't have a sense of how interesting the results are to the community.

**Claims And Evidence:**

Yes

**Claims Explanation:**

The paper clearly presents the idea of the method, describes the datasets used, and the experiments are cleanly designed. I have a couple of comments about the statistical tests use but I don't believe they change the conclusions of the paper. The paper also is clear about the limitations of the results.

**Requested Changes:**

For the analyses described in equations 1 and 2: what evidence is there that the data are sufficiently normal that this is a good way to analyze things? I wonder if there are non-parametric tests/bounds that are more appropriate here, e.g. something using KS like Dvoretsky-Kiefer-Wolfowitz (DKW) Inequality.

I wonder if the First Two Moments (FTM) terminology is the right description here --- mainly, do we need an opaque acronym? The literature is littered with so many acronyms, I wonder if there is a short name that is also accurate which doesn't need to be simplified. Just a thought!

---

### Review · Reviewer_2uJc · 2026-06-05

**Summary Of Contributions:**

This paper investigates whether very simple distributional features—the mean and standard deviation of a time series—can provide a strong and interpretable baseline for time-series classification, sometimes rivaling or outperforming more complex dynamic-feature models. The paper tests a baseline on 129 filtered UEA/UCR univariate time-series classification problems, excluding datasets incompatible with catch22 feature computation. For each dataset, the paper computes FTM features and compares them against a combined FTM + catch22 feature set, where catch22 represents 22 established time-series features capturing dynamics such as autocorrelation, outlier timing, periodicity, and long-range dependence. Performance is assessed using the standard UEA/UCR resampling protocol: the original split plus 29 stratified resamples, with train-only z-score normalization applied to features before testing. Accuracy is the main metric. Statistical comparisons include a one-sided chance-performance test for FTM and a corrected resampled t-test for comparing FTM with FTM + catch22. The paper also includes a neuroimaging case study using resting-state fMRI from 164 participants to classify schizophrenia versus controls using regional FTM and catch22 features with repeated stratified 10-fold cross-validation.

Overall, the paper has some interesting potential contributions. First, the paper highlights that complex time-series classifiers should be compared against simple baselines before claiming that temporal dynamics are necessary. Testing across 129 UEA/UCR datasets gives the claim much more weight than a single-domain demonstration

**Audience:**

Yes

**Audience Explanation:**

The contribution is significant as a benchmarking and interpretability warning.  However, the methodological contribution is moderately novel. Mean and standard deviation are not novel features, and simple baselines are not a new idea. The novelty lies in systematically showing, across a major time-series repository and an fMRI case study, that these trivial non-temporal features can explain surprisingly strong classification performance.

**Broader Impact Concerns:**

A discussion of broader impacts would be useful to add to the paper.

**Claims And Evidence:**

No

**Claims Explanation:**

The empirical design is reasonable as the authors use established benchmark datasets, train/test resampling, train-only normalization, linear SVMs, and corrected statistical testing for correlated resamples. The central argument is also sound: if mean and standard deviation perform well, then more complex classifiers may be exploiting distributional or scale differences rather than temporal dynamics.

However, the soundness is weaker in the neuroimaging case study because age, sex, motion, scanner/preprocessing effects, and group imbalance could partly account for the performance. The paper acknowledges some generalizability concerns, especially measurement-scale sensitivity, but stronger controls are needed before interpreting the fMRI result as biologically meaningful.

The paper identifies datasets where the mean and standard deviation are informative, but it does not provide a detailed categorization of whether this reflects meaningful signal, measurement artifacts, preprocessing inconsistencies, leakage-like calibration effects, or domain-valid class differences. Furthermore, there seem to be no adjustments applied to age and sex variables, which is necessary, since the schizophrenia and control groups differ in age and sex. So in summary, no theory or empirical criteria are provided on when using mean and stnadard deviation as predictors would provide better-than-random performance.

**Requested Changes:**

Would FTM still perform well after per-time-series z-scoring or other normalization that removes mean and variance differences?

For the fMRI case study, does FTM remain predictive after adjusting for age, sex, motion, and site/acquisition-related covariates?

How robust is FTM performance under rescaling, additive offsets, simulated sensor drift, or external validation splits?

---

### Review · Reviewer_Wkre · 2026-07-15

**Summary Of Contributions:**

This paper explores how well mean and standard deviation of a time-series can be used to perform classification. It evaluates this simple model on many tasks from a set of time-series classification problems plus a neuroimaging one. It also compares to models with features that rely on temporal ordering. They find that this simple baseline can perform above chance on many problems.

**Audience:**

Yes

**Audience Explanation:**

People doing certain kinds of time series classification should be aware of this strong baseline.

**Claims And Evidence:**

Yes

**Claims Explanation:**

Technically, yes. Figure 1 shows this simple classifier working better than chance on many problems.

**Requested Changes:**

I think the framing and conclusions of this paper need work. As various points throughout, previous work is referenced that appears to do something very similar to the current work (for example references Bryant et al., 2024, Mignan & Broccardo, 2019,  Fulcher et al., 2013, Wu et al., 2023). Before describing the results, the authors should clearly delineate what is novel about their work versus past work.


The conclusions could also be more thoroughly explored and stated. First, it would be good to contextualize these findings with respect to how more complex (deep net) models perform. Is the claim that this baseline is on par with deep learning models on most of these tasks? Or do the deep learning models still perform better, just not as much better as a comparison to chance performance would suggest?

Second, it would be good to understand the likelihood that these more complex models perform well because they have (indirect) access to mean and standard deviation. As the authors discuss, sometimes time-series data is normalize on a per sample basis, which would destroy this information. But sometimes it isn't. In studies that use deep nets on these problems to achieve good performance, is that mean and std information available or are is per-sample standardization used? This is important to understand especially because the authors note how in many cases mean and std differences depend on precise data collection methods that are unlikely to generalize (whereas true temporal trends may). So it matters if these more complex models are relying on mean and std (in which case they would be as brittle as the FTM model) versus if they are not (and therefore this is not really a fair baseline to compare them against). In general, the authors should do more work exploring the current literature in order to make their findings more interpretable and impactful. Especially given that this paper is otherwise light.

Is the fact that catch-22 + FTM performs worse than FTM due to overfiting? The authors should explicitly explore this by showing learning curves for train and test data. Does the combined model work better on training data? Were the regularization parameters explored to give each model the best chance?